# Report of the Influenza Vaccination Program in Mexico (2006–2022) and Proposals for Its Improvement

**DOI:** 10.3390/vaccines11111686

**Published:** 2023-11-03

**Authors:** Rodrigo Romero-Feregrino, Raúl Romero-Cabello, Mario Alfredo Rodríguez-León, Valeria Magali Rocha-Rocha, Raúl Romero-Feregrino, Berenice Muñoz-Cordero

**Affiliations:** 1Asociación Mexicana de Vacunología, Mexico City 06760, Mexico; 2Instituto Para el Desarrollo Integral de la Salud (IDISA), Mexico City 06700, Mexico; 3Employer Sector CONCAMIN, Technical Council, Instituto Mexicano del Seguro Social (IMSS), Mexico City 06600, Mexico; 4Saint Luke School of Medicine, Mexico City 11000, Mexico; 5Department of Infectology, Hospital General de México, Mexico City 06720, Mexico; 6Department of Microbiology and Parasitology, Faculty of Medicine, Universidad Nacional Autónoma de México (UNAM), Mexico City 04360, Mexico; 7Academia Mexicana de Pediatría, Mexico City 03810, Mexico; 8School of Higher Studies (F.E.S.) Zaragoza, Universidad Nacional Autónoma de México (UNAM), Mexico City 09230, Mexico; 9School of Life and Health Sciences, Universidad Popular Autónoma del Estado de Puebla, Puebla 72410, Mexico; 10Department of Pediatrics, Hospital General de Cuajimalpa IMSS-Bienestar, Mexico City 05230, Mexico

**Keywords:** influenza, influenza vaccine program, Mexico, low coverage, vaccine acquisition

## Abstract

Background: Influenza has continued to be an important public health challenge, and the WHO recommends that countries consider vaccination for persons at high risk. Mexico has such a program, and we sought to determine its current situation. Material and Methods: This study has an ecological, longitudinal, and retrospective design based on secondary information on the acquisition and application of vaccines against influenza from official data from 2006 to 2022. Results: We found annual variability in the numbers of purchases and application of doses, with coverage of less than 90% of the total population every year; in addition, 17 million vaccines were not used in this period. Discussion: This study shows the situation of the program at the national level. Two institutions acquired fewer the vaccines, while one purchased more for its target population, but the necessary vaccinations were not acquired. On average, 1.7 million of the vaccines purchased annually were not used, but in some years, more vaccines were applied than were purchased in all of the institutions. We also observed that, between institutions, the vaccine coverage was very different, from 21% to 180%. On average, 6.2 million people were not vaccinated annually, between 16% and 22% of the target population, demonstrating low coverage. When we compared the coverage data that we calculated to the data published by the institutions, a great difference was observed. Conclusions: We found inconsistencies in the data, indicating their unreliability and potential disorganization within the program, as the target populations of each institution were not clear. In addition, the application data may have had reporting errors. Adequate coverage was not achieved, and the coverage was different from that reported in the official sources. We propose the implementation of different systems for control, evaluation, and access to the information of the program.

## 1. Introduction

Universal vaccination programs have existed for several decades in Mexico. They have been based on different strategies and have effectively reduced, controlled, and eradicated preventable diseases through the administration of different biological materials.

However, these programs and strategies have deteriorated over time. In the last twenty years, there has been a deficiency in their coverage, which national and international organizations have determined as the cause of this deterioration.

In terms of coverage, a decrease has been observed in recent years in Mexico in the application of the complete vaccination schedule in children, from 96% in 2018 to 76% in 2019, 33% in 2020, and 86.6% in 2022, as reported by the Pan-American Health Organization (PAHO) and the National Health and Nutrition Survey (ENSANUT, Spanish acronym) 2021 for COVID-19. Similarly, a supply shortage of several vaccines was reported in 2019 [1,2]. The PAHO noted that in 2021, 1.7 million children under 1 year of age (14%) had not received a single dose of the basic vaccination schedule. More than 50% of these cases were concentrated in two countries, Mexico and Brazil. It is for this reason that the PAHO itself proposed a goal for Vaccination Week in the Americas in 2023: to reach more than 92 million people with more than 144 million doses of different vaccines in 45 countries [3].

Other recent studies described the vaccination coverage in children and adolescents in Mexico in 2022 and compared it to the observed prevalence of vaccine-preventable diseases with data derived from the ENSANUT 2021. According to the analysis, the coverage goal of 90% was not reached for any of the investigated immunogens; it was highlighted that the coverage for the first dose of measles, mumps, and rubeola (MMR) was markedly reduced [4,5].

Based on a complete review of different data in ten Latin American countries, Rombini et al. determined the rankings of immunization programs. As results, they showed that there was a seasonal flu vaccine program in all countries. In Argentina and Colombia, it is first applied to the population at between 6 and 24 months of age. Paraguay and Peru extended it up to 35 months, while Brazil, Chile, Costa Rica, Mexico, Panama, and Uruguay further extended it up to 59 months [6]. 

They concluded that the ranking was led by Chile and Panama with vaccinations in the first and second years of life. They were followed by Argentina, Uruguay, and Costa Rica, which stood out due to their vaccination of other groups with their anti-flu (influenza) vaccines and programs. Brazil, Colombia, and Mexico showed the most backward national vaccination schedules, programmatic gaps, and the lowest vaccination coverage [6].

It is important to highlight the benefits resulting from effective, timely, and safe immunization against influenza. It was not until the middle of the 20th century that the first vaccines were available for application in populations at higher risk [7]. In 2002, in the United States of America, the Advisory Committee on Immunization Practices (ACIP) started the application of the influenza vaccine to children aged 6 to 23 months [8]. 

In Mexico, in 2004, vaccination against seasonal influenza was introduced for children 6 to 23 months of age, and adults 65 years and older. In 2005, the age groups were expanded to 6 to 35 months, and in 2010, they were expanded to 6 to 59 months [9], and it has remained that way until 2022.

In 2009, a new strain of influenza virus (H1N1) caused a new pandemic that caused almost 300,000 hospitalizations and just over 12,000 deaths [7]. During only the 2019–2020 influenza season, the influenza vaccine prevented an estimated 7.52 million infections, 3.69 million medical visits, 105,000 hospitalizations, and 6300 deaths in the United States [10].

Influenza causes an estimated 444,000–553,000 deaths worldwide every year. Most of these deaths occur among elderly adults and persons with chronic cardiopulmonary disease, but the toll includes 28,000–111,000 deaths among children less than five years of age [11]. 

In view of this global disease burden, many countries have implemented influenza vaccination programs in order to reduce morbidity and mortality due to influenza. The World Health Organization (WHO) recommends that countries consider influenza vaccination for persons at high risk of death or serious illness from influenza, and it has recognized the following groups as being at increased risk: pregnant women, children aged 6–59 months, elderly adults, individuals with specific chronic medical conditions, and healthcare workers [11].

Despite the efforts applied, as well as global and national policies based on the prevention of influenza through vaccination in at-risk populations, influenza has continued to be an important public health challenge. Kamidani et al. analyzed hospitalizations due to influenza in the United States following the 2009 H1N1 pandemic. They highlighted that among 13,235 hospitalized children with influenza, 2676 (20%) were admitted to the ICU, 2262 (17%) had pneumonia, 690 (5%) required mechanical ventilation, and 72 (0.5%) died during hospitalization. Influenza type A was predominant among the hospitalized children in each season. Influenza B accounted for 24% of the total cases but ranged seasonally from 4% to 42% of all hospitalizations. The rate of influenza vaccination was highest among children from 6 months to 2 years of age (45%) [12].

Based on data showing a gradual decrease in vaccination coverage that was concomitant with the number of cases diagnosed in recent years, shortages in vaccine supplies, and different information published about influenza coverage, in addition to our knowledge on vaccinology and the Mexican health system, we considered that it is necessary to study the official data on acquisitions, application, and coverage in order to ascertain the actual situation of our country in terms of influenza vaccination. 

Our aim is to describe and compare the data obtained to understand the current situation of the vaccination program, to propose changes that address the issues found. The following questions arise: what is the real coverage? If we have low coverage, what are the causes? How can we improve this situation?

In this report, we expected to find a low agreement between the data reported by public health institutions and our own calculations, as there was great variability in the information and little concordance among the data. 

We would like the results of this report to constitute a basis for the generation of proposals that are aimed at urgently improving the vaccination program against influenza.

## 2. Material and Methods

A study with an ecological, longitudinal, and retrospective design was carried out based on secondary information on the official data from 2006 to 2022.

This study is of a descriptive and analytical nature. We focused on vaccine acquisition and application as reported by the three major government health institutions in Mexico, and we calculated the target populations and coverage. We analyzed and compared data on the numbers of doses purchased by each institution, as well as the information concerning the target populations and the numbers of applied doses of the influenza vaccine that were acquired within each year.

The major government health institutions in Mexico are the Mexican Social Security Institute (IMSS, Spanish acronym), the Security and Social Services Institute for State Workers (ISSSTE), and the Ministry of Health (SSA). The combined efforts of these three institutions address 98% of the medical care needs of the country, which has a population surpassing 128 million inhabitants and 2.2 million births per year, as reported in 2020 [13,14,15,16,17,18].

We established models to compare the numbers of vaccine doses acquired, their application rates, and the overall coverage.

The accuracy and reliability of this study depend on having abundant amounts of data from official sources. This information was used to establish suitable theoretical models for identifying trends and providing adequate explanations of the results. It was also important to obtain clear conclusions that were aimed at making proposals and suggesting actions for improvement.

This study was conducted based on the steps described in the following.

### 2.1. Information

We established the types of data required to conduct the study, i.e., beneficiaries per institution, vaccine description, the number of doses acquired, and the number of doses applied. All of the information on purchases was requested from and delivered by the databases of the National Institute for Transparency, Access to Information, and Protection of Personal Data (INAI, Spanish acronym) [19,20,21,22,23,24,25,26,27,28,29,30,31,32], and the information on dose application was requested and obtained from the databases for historic information of the IMSS [33], ISSSTE [34], and SSA [35].

### 2.2. Analysis

After the above information was obtained, we established models for comparing all of the data. The following terms were considered for data interpretation:Theoretical target population: the number of individuals that should be vaccinated based on the population considered by the respective institution and the indications for the vaccine;Percentage variation in annual acquisitions: percentage variation in annual purchases after comparing a particular year with the next one;%PUR: percentage of vaccines acquired with respect to the theoretical target population;%APP: percentage of vaccines applied with respect to those acquired;%COV: percentage of vaccines applied with respect to the theoretical target population.

The theoretical target population was calculated to identify the necessary number of vaccines for a particular target population based on the population of the data obtained from the National Population Council (CONAPO) database and based on the indications for the vaccine according to Mexico’s national vaccination scheme.

The official vaccine indications are for persons who are <5 years of age, >60 years of age, or pregnant, as well as populations aged from 5 to 59 years with comorbidities such as diabetes mellitus (Table 1), congenital heart or lung diseases, morbid obesity (BMI > 40), chronic lung disease, including COPD and asthma, cardiovascular disease (except for essential arterial hypertension), chronic kidney disease, immunosuppression acquired through illness or treatment, cancer, or HIV/AIDS [9].

In this case, we calculated this value only with the data of persons who are <5 years of age, >60 years of age, pregnant, and 18 to 59 years with diabetes. For persons with diabetes mellitus, we used the percentage of the population with the disease as reported in the ENSANUT. 

The population aged from 5 to 59 years with comorbidities, such as congenital heart or lung diseases; morbid obesity (BMI > 40); chronic lung disease, including COPD and asthma; cardiovascular disease (except for essential arterial hypertension); chronic kidney disease; immunosuppression acquired through illness or treatment; cancer; or HIV/AIDS, was not calculated due to the lack of adequate data.

The theoretical target population for each institution was calculated by using a formula that depended on the total target population and the percentage of the beneficiary population of each institution, i.e., the IMSS, ISSSTE, and SSA. Table 2 shows the respective percentage per institution of the beneficiary population as annually reported. 

Based on these data, we calculated the target population percentage for each institution [13,14,15,16,17,18], the formulas are the following:Total theoretical target population: population <5 years of age, >60 years of age, pregnant, and 18 to 59 years with diabetes;IMSS: Total theoretical target population * IMSS Percentage of the population per year;ISSSTE: Total theoretical target population * ISSSTE Percentage of the population per year;SSA: Total theoretical target population * SSA Percentage of the population per year.

The percentage variation in the analysis of annual purchases and the number of doses annually acquired for each institution were evaluated by using the following expression:

([Acquired number − number acquired in the previous year]/number acquired in the previous year) ∗ 100

A comparison was performed regarding the acquisition percentage for a particular target population and the percentage of vaccine application, where

%PUR = (amount of acquired doses/theoretical target population) ∗ 100;

%APP = (number of applied doses/amount of acquired doses) ∗ 100;

%COV = (number of applied doses/theoretical target population) ∗ 100.

The data were processed in GraphPad Prism version 9.3.0 for Windows (GraphPad Software, www.graphpad.com, accessed on 24 July 2023). The figures were prepared with different combinations of data. Only representative data are shown. 

Finally, we calculated by institution and added the number of unvaccinated people with the theoretical target population formula minus vaccines applied.

## 3. Results

The data for each year were located in the official sources for the doses acquired and doses applied. We found the following amount of data for each institution, where each number represents the years for which we found data, and the percentage found is in parentheses. The maximum is 17 datapoints, that is, one per year, which represents 100%:Doses acquired annually: 10 for SSA (59%), 17 for ISSSTE (100%), and 11 for IMSS (67%). Most of the information was successfully retrieved, except for that of the IMSS in the period of 2006–2011 and that of the SSA in 2006–2012.Doses applied annually: 17 for SSA (100%), 16 for ISSSTE (94%), and 13 for IMSS (76%). Most of the information was successfully retrieved, except for that of the IMSS in the period of 2006–2009 and that of the ISSSTE in 2006.

### 3.1. Acquired Doses

Figure 1 shows the numbers of influenza doses acquired and the percentage variation in annual acquisitions. A comparison was made between the annual acquisitions of each institution and the overall number (Figure 1).

The annual acquisitions were highly variable in terms of the numbers of vaccines purchased (Figure 1). The average, maximum, and minimum values for each institution were, respectively, as follows: 5.08%, 26.64%, and −16.86% for the IMSS; 19.57%, 287.87%, and −20.13% for the ISSSTE; and −3.67%, 31.69%, and −11.26% for the SSA. However, the values for the total numbers of doses were 4.11%, 22.28%, and −7.10%, respectively. For example, for the ISSSTE, the variation in 2007 was 287.82%, which meant that there was an increase from 500,000 to 1,939,360 doses, and in 2014, it was 64.28%, which meant that there was an increase from 1,400,000 to 2,299,900 doses, and for the IMSS, the variation in 2018 was −16.86%, which meant that there was a decrease from 12,885,200 to 10,712,700 doses.

The first year of the ISSSTE (from 2006 to 2007) is represented with a line outside the graph because the percentage variation was 287.82%; if placed on the correct scale, the graph would be very large and difficult to read.

### 3.2. Application and Coverage

Figure 2 shows the results of the annual administration of influenza vaccines for according to institution and in total.

Figure 2 shows that the number of vaccines administered is not constant each year and that it varies significantly in each institution and in the total.

Figure 3 shows the results of the percentages of PUR, APP, and COV.

Figure 3 shows that, for the IMSS, from 2012 to 2022, %PUR was lower than 80% in all years, %APP was lower than 90% in all years (except in 2018, when it was 127%), and %COV was 70% or lower from 2010 to 2022. For the ISSSTE, from 2006 to 2022, %PUR was lower than 70% in all years, %APP was very variable from 31% to 127% in the period from 2007 to 2022 (in 8 years, more than 100% of the purchased vaccines were applied), and %COV was lower than 60% in all of the years from 2007 to 2022. For the SSA, from 2013 to 2022, %PUR was greater than 120% in all years (except in 2013, when it was 95%), %APP was between 53% and 122% (in 7 years, more than 100% of the purchased vaccines were applied), and %COV was between 44% and 163% from 2006 to 2022 (in 9 years, it was greater than 100%). Finally, considering the three institutions together, from 2013 to 2022, %PUR was between 77% and 99%, and %APP was between 77% and 99% (in 3 years, more than 100% of the purchased vaccines were applied). From 2010 to 2022, %COV was between 68% and 99%.

Figure 4 presents a comparison of the influenza vaccine purchases, the theoretical target populations, and the application rates by institution and in total.

Figure 4 shows a comparison of influenza vaccine purchases (Pur), the theoretical target populations (Obj), the application rates, and the medians of the target populations. The figures show that for the IMSS and ISSSTE, the numbers of vaccines that were purchased and applied were smaller than the target population. For SSA, from 2013 on, the numbers of vaccines that were purchased and applied were greater than the target population, except for 2015, when fewer were applied. In total, it was observed that the numbers of vaccines that were purchased and applied every year were smaller than the target population.

The institutions’ coverage according to the target population and application of vaccines was evaluated for each institution and in total, that is, with the three institutions together. We observed that for the ISSSTE and IMSS, most of the coverage was low, and for the SSA, after 2013, it was greater than 100% according to its theoretical population. In total, there was 90% coverage in only 4 years.

A statistically significant difference was found in the median number of vaccines purchased (*p* = 0.0020) and the number of vaccines applied (*p* = 0.0002) with respect to the median of the theoretical target population for the IMSS. 

A statistically significant difference was found in the median number of vaccines purchased (*p* < 0.0001) and the number of vaccines applied (*p* < 0.0001) with respect to the median of the theoretical target population for the ISSSTE. 

Although a statistically significant difference was found in the median number of vaccines purchased (*p* = 0.0039) with respect to the median of the theoretical target population, this difference was not significant for the number of vaccines applied (*p* = 0.6441) for the SSA.

No statistically significant differences were found in the median numbers of total vaccines purchased (*p* = 0.3223) or total vaccines administered (*p* = 0.0840) with respect to the median of the total theoretical target population.

Figure 5 shows the results by institution and in total for the number of unvaccinated people per year. In the unvaccinated line, the negative numbers indicate unvaccinated people, while positive numbers represent people who were vaccinated beyond the proposed theoretical target. Finally, the calculated %COV is also shown.

It can be observed in Figure 5 that for the IMSS and ISSSTE and in total, there were unvaccinated people. For the three institutions together, their number ranged from 361 thousand in 2020 to 10 million in 2015. For the SSA, it was observed that since 2016, more people were vaccinated than there were in the theoretical target population.

The total number of vaccines that were not applied was calculated by subtracting the number of vaccines that were applied from those that were purchased. According to the results, 14% (19,841,462) of the vaccines were not applied by the IMSS from 2012 to 2022, 8% (2,481,949) were not applied by the ISSSTE from 2007 to 2022, and −0.03% (−431,121) were not applied by the SSA from 2013 to 2022. The total from 2013 to 2022 was 5.5% (17,191,292).

## 4. Discussion

This study shows the situation of the influenza vaccination program. It has limitations due to the amount of information because data were not found for all of the years that were searched. In addition, the total target populations were not completely calculated; only persons aged <5 years and >60 years, those who were pregnant, and those aged 18 to 59 years with diabetes were considered because we did not find reliable information on the number of people with comorbidities by year.

Despite these limitations, the amount of information found on the numbers of vaccines that were purchased and applied allowed this study to be carried out. It was possible to perform different calculations and an analysis of different periods for each institution and the total of the three institutions in order to gain an overview of the influenza vaccination program in Mexico.

We observed decreases and increases in the numbers of purchased vaccines that had no logical explanation. Theoretically, they should be constant, since the target population of each institution did not vary much annually. With the data, we were unable to identify the causes of these variations in all of the institutions.

We also reported the acquisitions according to the theoretical target populations, where we observed that two institutions (the IMSS and ISSSTE) acquired 80% or less of the vaccines needed for the target population in all years, and one institution (the SSA) acquired 120% or more. In total, in all years, the necessary vaccines were not acquired.

We also reported the percentages of vaccine application according to the acquisitions. We observed for some years that in all institutions, more vaccines were applied than were purchased, and the same trends were found in the total. 

Considering the three institutions together, on average, 1.7 million of the vaccines purchased annually were not applied, with a maximum of 10.4 million in 2015. In addition, 4.3 million excess vaccines were applied in 2018. In some cases, such as in 2017, this could be explained by the application of a half dose for children under 3 years of age to whom the vaccine was being applied for the first time, but in the other cases, the number was too large to have a logical explanation.

We also observed that, among the institutions, the vaccine coverage was very different. For the SSA, in the last 9 years, it was greater than 100%, and in the other institutions, it was less than 80%. The ISSSTE was the institution with the lowest coverage. 

At the national level, the total numbers of vaccines that were purchased and applied apparently coincided with the total theoretical target population; however, the analysis of each institution showed that in the cases of the IMSS and ISSSTE, the numbers of vaccines that were purchased and applied were much smaller than the theoretical target population. In addition, although the SSA apparently complied with the application of vaccines for the target population, it was observed that the number of vaccines acquired was much greater than the theoretical target population.

It is possible that the SSA sought to compensate for the lack of purchases and number of vaccines applied by the other institutions, but in total, we continued to observe coverage of less than 90% with a high annual variation.

The low coverage also showed that there were millions of people without vaccines. According to the target populations that were calculated, on average, 6.2 million people were not vaccinated annually from 2010 to 2022, with a maximum of 10.7 million in 2015 and a minimum of 361 thousand in 2020; this coincided with the COVID-19 pandemic, which likely increased interest in vaccination.

By comparing all of the data, we observed that there may be problems with reporting and data quality because there were inconsistencies in the information found in different sources. There were lack of uniformity and agreement among data sets depending on the source.

## 5. Conclusions

The study achieved its objective of gaining a general overview of the influenza vaccination program in Mexico, and we found irregularities in the data, which could indicate their unreliability and disorganization in the program, such as problems in determining the target population, in administration and purchases, in the execution of the program when applying the vaccines, or in the quality of the reported data.

According to the purchases and the application rates, the target population of each institution was not clear because they did not acquire the numbers of vaccines necessary for their theoretical target populations, which were calculated according to the beneficiaries reported by each one.

The application data may have contained errors because the application rate exceeded the numbers of vaccines purchased in many years, and this did not have an adequate explanation, except that it was a reporting error. In other years, there was a large amount of waste because not all of the purchased vaccines were applied.

The variation in the quantities of vaccines purchased has no explanation, and the vaccination coverage reported by the official sources was different from the data calculated in this study.

Even though the entire target population was not calculated according to the indications of the vaccination program in Mexico, adequate coverage was not achieved.

It is necessary to reorganize the vaccination program against influenza as a country and for each institution, with true political and operational willpower, in order to increase the benefits of the program. All institutions must work together to review beneficiaries’ nominal information to calculate the real target populations and be active in providing influenza vaccinations to the population.

We propose that each institution identifies its target populations, buys the necessary vaccines for each year, and implements an electronic system for controlling the application of vaccines, which is constantly reviewed. A purchase control system that can be audited and constantly reviewed is also necessary. In addition, all government vaccine information should be public and easily accessible, with data on acquisitions, coverage, rejections, shortages, and more.

## Figures and Tables

**Figure 1 vaccines-11-01686-f001:**
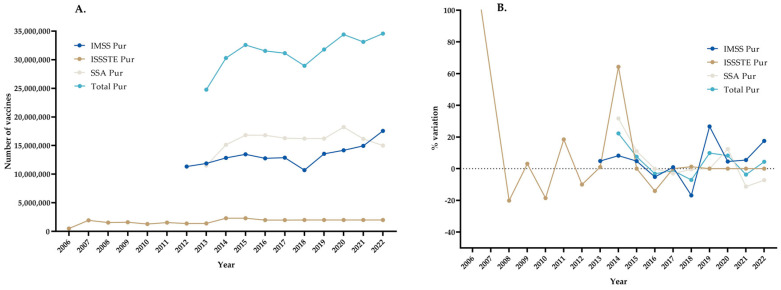
Influenza vaccines: Annual acquisitions and percentage variation in annual acquisitions. (**A**) Annual Acquisitions, (**B**) % variation in annual acquisitions.

**Figure 2 vaccines-11-01686-f002:**
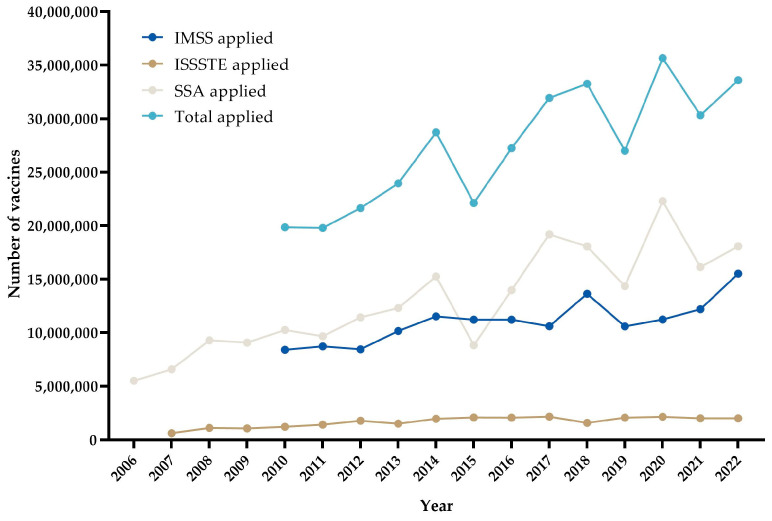
Influenza vaccines: annual application.

**Figure 3 vaccines-11-01686-f003:**
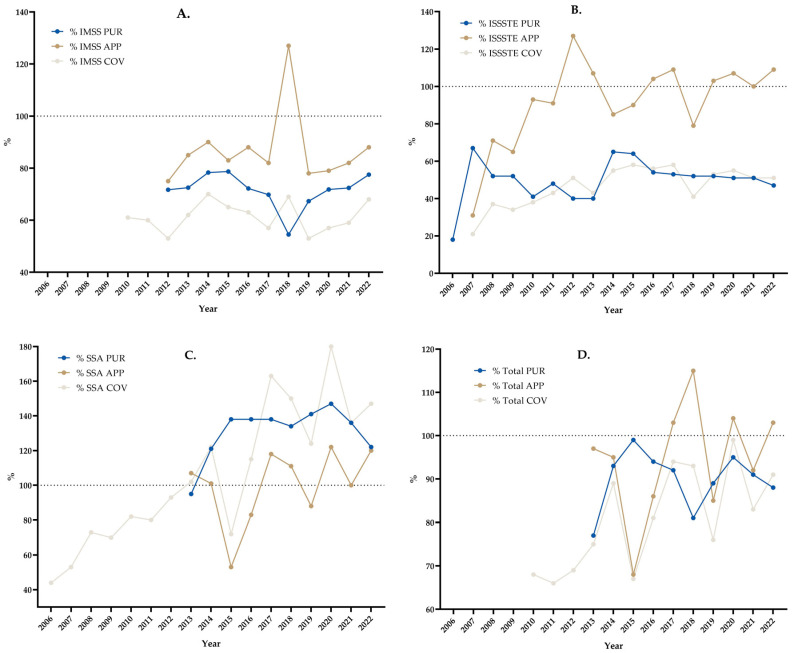
Influenza vaccine percentage of the amount of acquired doses according to theoretical target population (PUR), of the number of applied doses according to amount of acquired doses (APP) and vaccine coverage (COV). (**A**) % IMSS, (**B**) % ISSSTE, (**C**) % SSA, (**D**) % Total.

**Figure 4 vaccines-11-01686-f004:**
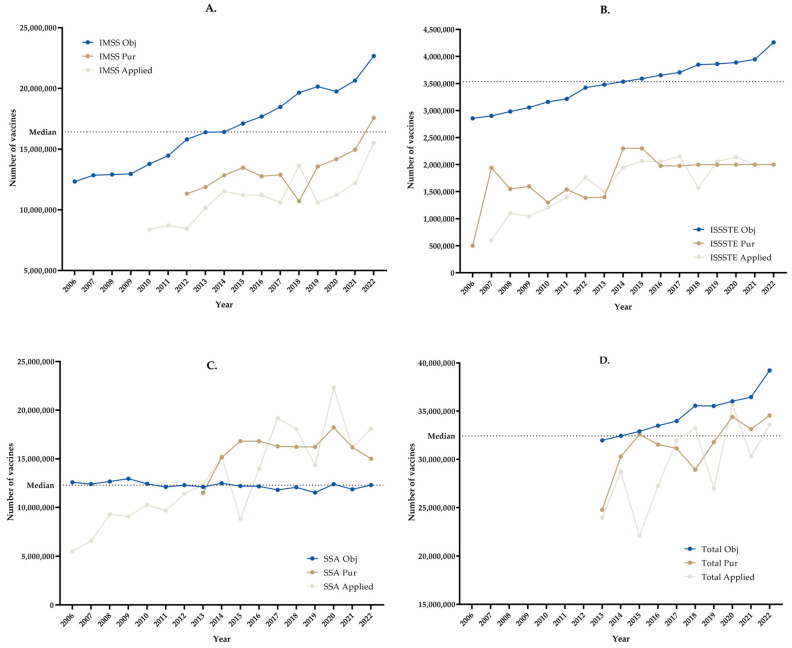
Influenza vaccines: comparison of influenza vaccine purchases, theoretical target populations, and application rates. (**A**) IMSS, (**B**) ISSSTE, (**C**) SSA, (**D**) Total.

**Figure 5 vaccines-11-01686-f005:**
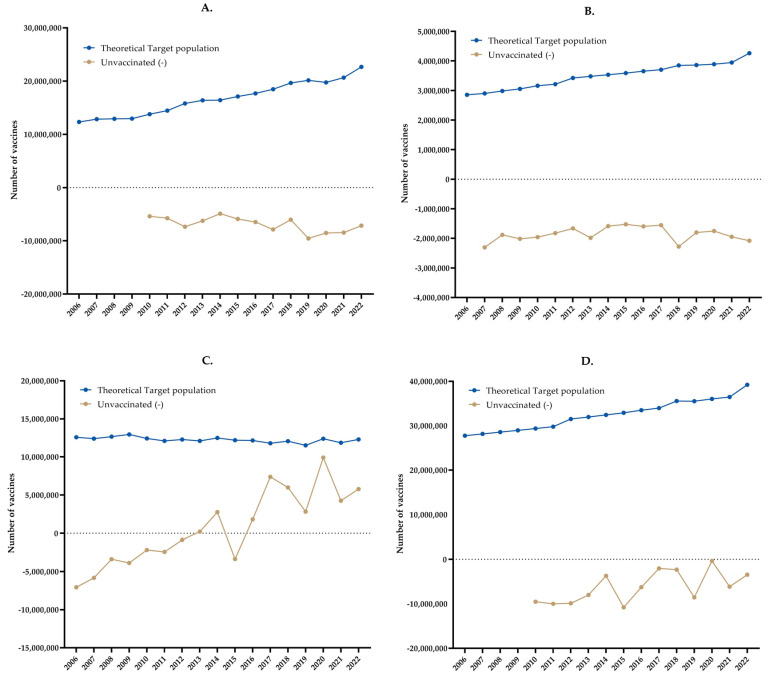
Influenza vaccine coverage and proportion of unvaccinated population. Mexico, 2006–2022. (**A**) IMSS, (**B**) ISSSTE, (**C**) SSA, (**D**) Total.

**Table 1 vaccines-11-01686-t001:** Percentage of population reported with diabetes.

Years	2006–2011	2012–2017	2018–2020	2021	2022
Diabetes	7% [35]	9.2% * [36]	10.3% * [37]	10.2% * [38]	12.6% * [39]

* Percentage of population reported to have diabetes in the different iterations of the ENSANUT [36,37,38,39,40].

**Table 2 vaccines-11-01686-t002:** Percentage of the population per institution per year (%).

	2006	2007	2008	2009	2010	2011	2012	2013	2014	2015	2016	2017	2018	2019	2020	2021	2022
IMSS	43	44	44	44	46	47	49	50	50	51	52	54	55	56	54	56	57
ISSSTE	10	10	10	10	10	11	11	11	11	11	11	11	11	11	11	11	11
Other	2	2	2	2	2	2	2	2	1	2	2	1	1	2	2	2	2
SSA	45	44	44	44	42	40	38	37	38	36	35	34	33	31	33	31	30

## Data Availability

The data are available from sources of each institution cited in the references.

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
