# Peer review of "Report of the Influenza Vaccination Program in Mexico (2006–2022) and Proposals for Its Improvement"

_vaccines, 2023, doi:10.3390/vaccines11111686_

Round 1
Reviewer 1 Report (Previous Reviewer 2)
Comments and Suggestions for Authors
The revised version of the article “The challenges of influenza vaccination in Mexico (2006–2022):Review of 17 years of the program and proposals for its improvement” has been reviewed 5
21
ABSTRACT:
· On average,1.7 million of the vaccines purchased annually were not used, but in some years, more vaccines were applied than were purchased in all of the institutions.
How can you apply more vaccines than those you have purchased?
· We also observed that, between institutions, the vaccine coverage was very different from 21% 32 to 180%.
180% ??? please confirm this
Line 182 on pg 4
In this case, we calculated this value only with the data ofpersons who are <5 years of
No need to repeat Diabetes at each one
Table 1. Percentage of population reported with diabetes.
Years |
2006-2011 |
2012-2017 |
2018-2020 |
2021 |
2022 |
Diabetes |
Diabetes 7% [35] |
Diabetes 9.2%* [36] |
Diabetes 10.3%* [37] |
Diabetes 10.2%* [38] |
Diabetes 12.6%* [39] |
Pg 5 Line 223
Finally, we calculated by institution and total the number of unvaccinated people with …
Change Figure 2 and 3 , 5 captions:
Figure 2 show the results of the Annual administration of influenza vaccines according to institution and in total.
Figure 3 .Influenza vaccine percentage of the amount of acquired doses according to theoretical target population (PUR), of the number of applied doses according to amount of acquired doses) (APP) and vaccine coverage ( COV).
Figure 5. Influenza vaccine coverage and proportion of unvaccinated population. Mexico 2006-2022
Comments on the Quality of English LanguageAccetable with minor corrections
Author Response
ABSTRACT:
- On average,1.7 million of the vaccines purchased annually were not used, but in some years, more vaccines were applied than were purchased in all of the institutions.
How can you apply more vaccines than those you have purchased?
RRF: Yes, it is one of the findings that suggests problems in government records and data quality.
- We also observed that, between institutions, the vaccine coverage was very different from 21%32 to 180%.
180% ??? please confirm this
RRF: Confirmed that is the correct data of 180%.
Line 182 on pg 4
In this case, we calculated this value only with the data ofpersons who are <5 years of
No need to repeat Diabetes at each one
Table 1. Percentage of population reported with diabetes.
Years |
2006-2011 |
2012-2017 |
2018-2020 |
2021 |
2022 |
Diabetes |
Diabetes 7% [35] |
Diabetes 9.2%* [36] |
Diabetes10.3%* [37] |
Diabetes10.2%* [38] |
Diabetes 12.6%* [39] |
Pg 5 Line 223
Finally, we calculated by institution and total the number of unvaccinated people with …
Change Figure 2 and 3 , 5 captions:
Figure 2 show the results of the Annual administration of influenza vaccines according to institution and in total.
Figure 3 .Influenza vaccine percentage of the amount of acquired doses according to theoretical target population (PUR), of the number of applied doses according to amount of acquired doses) (APP) and vaccine coverage ( COV).
Figure 5. Influenza vaccine coverage and proportion of unvaccinated population. Mexico 2006-2022
RRF: Recommended changes were made

Reviewer 2 Report (Previous Reviewer 3)
Comments and Suggestions for Authors
The observations were considered by the authors
Comments on the Quality of English LanguageAcceptable
Author Response
There is no correction requested
This manuscript is a resubmission of an earlier submission. The following is a list of the peer review reports and author responses from that submission.
Round 1
Reviewer 1 Report
Comments and Suggestions for Authors
The challenges of influenza vaccination in Mexico (2006––2022): Review of 17 years of the program and proposals for its improvement
In this study Romero-Feregrino et al. evaluated the acquisition, application, and coverage of influenza vaccines in three health care institutions from Mexico. The authors found that on average a significant amount of the vaccines purchased were not used, but in some years, more vaccines were applied than
were purchased in all the institutions. The authors suggest that these irregularities could be associated with confusions within the vaccination program, such as an unclear definition of the target populations or data reporting errors. The authors propose that different systems for control and evaluation should be implemented to improve the vaccination program.
The discussion of the results present on this work is poor and the authors present data in a very confusing manner, complicating the reading. The description of the figures and tables is not clear and not well organized. In addition, it is difficult to comprehend how the different values used for the analysis were calculated. Especially, the description of how the theoretical target population was calculated is confusing. This is a critical issue, because this parameter is essential to understanding the conclusions made by the authors. This manuscript would require substantial changes before acceptance for publication.
Major Comments
1. The title of the paper should be reconsidered. According to the title, the work is focused on the challenges of influenza vaccination based on a review of 17 years of the vaccination program. However, the authors only present data about the purchase, application, and coverage of the vaccine. A review of the program should also consider how other aspects of the vaccination program have changed over time. Such as policies for the implementation of programs, distribution logistics, communication tools used to disseminate immunization and vaccine information, influenza surveillance, etc.
2. In lines 50 to 53, the authors describe that the coverage of vaccination decreased from 96% in 2018 to 76% in 2019, 33% in 2020, and 86.6% in 2022 as reported by the Pan-American Health Organization (PAHO) and the National Health and Nutrition Survey (ENSANUT, Spanish acronym) 2021 for COVID19. However, it is unclear which vaccine they are referring to. Is it COVID19 vaccines or influenza?
3. In lines 54 and 55, the authors mention that in 2021 the PAHO reported that 1.7 million children under 1 year of age (14%) had not received a single dose of a vaccine. However, is not clear about which vaccine coverage they are talking about.
4. According to the definition of theoretical target population (lines 160-162), this term describes how many individuals should be vaccinated based on the population considered by each institution and the indications for the vaccine. However, in the lines 191 to 195 the authors describe that the theoretical target population for each institution was calculated by using a formula that depends on the necessary number of vaccine doses per individual institution. And that this formula considered the theoretical number of doses required and the respective percentage of the population using them as annually reported by each institution. This description is very confusing, why the value of the theoretical target population depends on the necessary number of vaccine doses? This seems counterintuitive. To know the necessary number of vaccine doses for each institution, we need to know the value of the theoretical target population. In addition, the authors mention that they also consider the theoretical number of doses required and the respective percentage of the population using them. How do they calculate these two parameters? And how these two values were used to calculate the theoretical target population? The authors should describe as accurately as possible how this calculation was made. Also, they should present the actual formula used and describe with precision all parameters involved.
5. According to the title of table 1, this table shows the percentage of population reported with diabetes. However, columns 2-4 of table 1 did not show any data related to diabetes. In addition, the legend of the table does not describe how this percentages were calculated. Are these percentages based on the theoretical target population of all institutions or the total population of Mexico? Please correct this table.
6. In lines 194 and 195 the authors describe that table 2 shows the respective percentage of the population using the vaccines. However, the title of table 2 is “Percentage of the population per institution per year”. This is confusing. What data is presented in table 2?
7. In lines 215 to 220, the authors present the data for the doses acquire and applied annually by each institution. However, it is not defined if the numbers shown are the total or average of doses or if the values are in millions of doses or thousands of doses. Authors should clarify what these values are.
8. If the numbers shown in lines 215-217, represent the average millions of doses acquired annually by each institution, then, this data contradicts what it is shown on the left of Figure 1. According to line 215, the doses acquired annually were 10 million for SSA, and 17 million for ISSSTE. However, in the left graph of figure 1 we can observe that the numbers of doses acquire by SSA were above 10 million, while for ISSSTE the values were less than 5 million for all years. What is the reason for this difference?
9. Like above, for the doses applied annually (lines 218-220). If the numbers represent the average millions of doses applied annually by each institution. Then according to line 218 there was an average of 16 million of doses applied by ISSSTE. However, if we analyzed figure 2 and figure 4, the number of applied doses for ISSSTE was 2 million or less for all years. What is the reason for this difference?
10. In lines 228 to 230, the authors explain that the average, maximum, and minimum values of the percentage variation of annual purchases for the ISSTE were 19.57%, 287.87%, and -20.13% respectively. However, maximum value of 287.87% does not appear in figure 1 (right graph). The authors describe that this value was not added to figure 1 because it was difficult to understand the meaning of this value for the analysis (lines 235-236). If the authors decide to not discard this value for the analysis, then they should not consider the value 287.87% as the maximum for ISSSTE.
11. Figure 2 is only mentioned on lines 240 and 241. However, there is no discussion about this figure in the main text. What is the reason of showing this graph?
12. In lines 291 to 294, the authors describe that figure 5 shows the number of unvaccinated people per year. And that in this figure the negative numbers indicate unvaccinated people, while positive numbers represent people who were vaccinated beyond the proposed theoretical target. However, according to the legend of figure 5, the positive numbers of this figure represent the theoretical target population. What data is presented in figure 5? Why are unvaccinated people presented as negative numbers? How were the unvaccinated people calculated? How the authors calculate the people who were vaccinated beyond the proposed theoretical target?
13. Between lines 356 and 357, the manuscript shows a table with no title or description. This complicates the analysis of the data presented.
Minor changes
1. Figure 1, Figure 3, Figure 4, and Figure 5 contain more than one graph. However, since there is no other subclassification between the different graphs presented in each figure, it is difficult to understand which graph is described in the main text. Authors should add a subclassification to label each individual graph within the figures.
2. There is no description for the acronym SRP on line 64. What SRP stands for?
3. The format of references is not homogeneous. Please correct.
4. References 3, 11, and 40 are in Spanish. Please correct.
Comments on the Quality of English LanguageCould be improved
Reviewer 2 Report
Comments and Suggestions for Authors
Article The article “The challenges of influenza vaccination in Mexico (2006–2022):Review of 17 years of the program and proposals for its improvement” 5
21
ABSTRACT:
The content of the results section of the abstract is not informative and not precise. What do the authors mean by less than 90& of the population? The specific mean or yearly coverage should be informed . 17 million vaccines not used or 1.7 million and of how many ?
This study shows the situation of the program : What program? On national level or is it considered for different institutions?
vaccine coverage was very different amog institutions , but to what extent, was it significant?
On average, 6.2 million people were not vaccinated annually ( what proportion of the population is this? And for the age distribution?
*Sensitivity analysis to assess the performance of the program would be a good evaluation
Introduction:
The Universal vaccination refers to influenza immunization also? If the focus is on Influenza vaccine , why introduce to such an extent the rest of vaccines ?
Influenza vaccine recommendation for 2010, they were expanded to 6 to 59 months, is this stil so? What is the 2010-2022 policy?
This paragraph is more suitable for the Discussion:
Kamidani et al. analyzed hospitalizations due |
to influenza in the United States following the 2009 H1N1 pandemic. They highlighted |
that, among 13,235 hospitalized children with influenza, 2,676 (20%) were admitted to the |
ICU, 2,262 (17%) had pneumonia, 690 (5%) required mechanical ventilation, and 72 (0.5%) |
died during hospitalization. Influenza type A was predominant among the hospitalized |
children in each season. Influenza B accounted for 24% of the total cases but ranged sea- |
sonally from 4% to 42% of all hospitalizations. The rate of influenza vaccination was high- |
est among children 6 months to 2 years of age (45%) [12]. |
In giving results for statistical significance clarity please include 95% Cis
Tables would work better by changing rows to columns.
Influenza immunization is known to be below recommended standards worldwide, to say that when the authors inform of a 90% coverage it is difficult to believe, unless the concept is a different one . The content in itself is very unspecific and unclear. Improving language might help in making the work more understandable and comprehensive .
In summary this is more an evidence of lack of coordination of the national program in order to achive a cost-effective vaccination and prevention of seasonal influenza.
Comments on the Quality of English LanguageThe content in itself is very unspecific and unclear. Improving language might help in making the work more understandable and comprehensive .
Reviewer 3 Report
Comments and Suggestions for Authors
The authors sought to analyze the current situation of Mexican immunization program against influenza. They focused on vaccine application and acquisition using reported data by three major government health institutions in Mexico: the Mexican Social Security Institute (IMSS, Spanish acronym), the Security and Social Services Institute for State Workers (ISSSTE), and the Ministry of Health (SSA). Next, they calculated the target population and coverage for each year. They obtained the number of beneficiaries per institution, vaccine description, number of doses acquired, and number of doses applied. All of information on purchases was requested from and delivered by the databases of the National Institute for Transparency, Access to Information, and Protection of Personal Data (INAI, Spanish acronym). A set of assumptions was adopted for analyzing and comparing data. As the estimated values refer to the reference population instead of a sample, it is not clear why the authors used Wilcoxon signed-rank test for comparing the number of vaccinations purchased and those applied. If the authors aim to compare a determined value that express a trend measure over the years (average rate, median etc) they have to take into consideration the temporal dependency of the values related to each year. In this case, the procedure to be adopted must consider serial autocorrelation, resulting from temporal dependence. The time series analysis proposed by Prais-Wintsten would be a good option. Thus the results described between line 278 and 290 should be revised. The non-numbered Table in the Discussion should be removed as it has data that are not comparable. One alternative it would be the authors calculate the values for elderly population. I am afraid that the authors should adjust the study’s objective. The results describe the estimated values for vaccines purchased and applied annually. These values were compared to target population for each year. To produce an analyze of the current situation would require to gather data on planning and executing campaigns and programs. The study has merit to be published after revising these points and adjusting the objective. So the produced information can support the recommendations proposed by the authors. It is important to implement an electronic system for controlling the application of vaccines and an electronic system for controlling purchases. Both they must be audited and constantly reviewed. Information on immunization program should be public and easily accessible, with data on acquisitions, coverage, rejections, shortages, and more.
Comments on the Quality of English LanguageI am afraid that some editing of English language is required